# Personalised antimicrobial susceptibility testing with clinical prediction modelling informs appropriate antibiotic use

Alex Howard [1,2] ✉, David M. Hughes[3], Peter L. Green [4,5], Anoop Velluva[1], Alessandro Gerada [1,2], Simon Maskell [6], Iain E. Buchan [4,7,8] & William Hope [1,2,8]

Antimicrobial susceptibility testing is a key weapon against antimicrobial resistance. Diagnostic microbiology laboratories use one-size-fits-all testing approaches that are often imprecise, inefficient, and inequitable. Here, we report a personalised approach that adapts laboratory testing for urinary tract infection to maximise the number of appropriate treatment options for each patient. We develop and assess susceptibility prediction models for 12 antibiotics on real-world healthcare data using an individual-level simulation study. When combined with decision thresholds that prioritise selection of World Health Organisation Access category antibiotics (those least likely to induce antimicrobial resistance), the personalised approach delivers more susceptible results (results that encourage prescription of that antibiotic) per specimen for Access category antibiotics than a standard testing approach, without compromising provision of susceptible results overall. Here, we show that personalised antimicrobial susceptibility testing could help tackle antimicrobial resistance by safely providing more Access category antibiotic treatment options to clinicians managing urinary tract infection.

Antimicrobial resistance (AMR) is a major public health challenge. Effective antimicrobial therapy is a balance between optimising individual treatment outcomes and minimising selection pressure that accelerates the generation and spread of AMR[1]. The World Health Organisation (WHO) Access, Watch, Reserve (AWaRe) classification is a widely used framework to promote sustainable antimicrobial use[2]. In 2024, the United Nations General Assembly has set a target for at least 70% of global human antimicrobial use to be from the AWaRe Access category[3]. A key tool to help achieve this target is antimicrobial susceptibility testing (AST)[4].

AST tests a panel of antimicrobial drugs to assess potential antimicrobial activity against a pathogen (e.g., bacterium or fungus) for the purpose of directing antimicrobial treatment—it does this by exposing a pathogen of interest (retrieved from a patient) to a known

[1]Department of Clinical Pharmacology and Therapeutics, Institute of Systems, Molecular and Integrative Biology, William Henry Duncan Building, 6 West Derby Street, University of Liverpool, Liverpool L7 8TX, UK. [2]Liverpool University Hospitals NHS Foundation Trust, Mount Vernon Street, Liverpool L7 8YE, UK. [3]Department of Health Data Science, Institute of Population Health, University of Liverpool, Waterhouse Building Block B, Brownlow Street, Liverpool L69 3GF, UK. [4]Civic Health Innovation Labs, University of Liverpool, Liverpool Science Park, 131 Mount Pleasant, Liverpool L3 5TF, UK. [5]Department of Mechanical and Aerospace Engineering, School of Engineering, University of Liverpool, The Quadrangle, Brownlow Hill, L69 3GH Liverpool, UK. [6]Department of Electrical Engineering and Electronics, School of Electrical Engineering, Electronics, and Computer Science, University of Liverpool, The Quadrangle, Brownlow Hill, L69 3GH Liverpool, UK. [7]Department of Public Health, Policy & Systems, Institute of Population Health, University of Liverpool, Waterhouse Building Block B, Brownlow Street, Liverpool L69 3GF, UK. [8]These authors jointly supervised this work: Iain E. Buchan, William Hope. ✉e-mail: alexander.howard@liverpool.ac.uk

antimicrobial concentration in a disc, testing medium, or agar. Disc methods measure antimicrobial susceptibility using the zone of inhibition size around an antimicrobial disc, typically testing six or eight antimicrobial agents per specimen. The minimum antimicrobial concentration that prevents microbial growth (minimum inhibitory concentration [MIC]) can be measured by broth dilution methods or agar methods (e.g., Etests), testing a variable number of antimicrobial agents per specimen depending on the method and testing platform used. The zone of inhibition size or MIC are used to determine whether a particular antimicrobial agent can be used to treat an infection caused by the pathogen, using cutoff values provided by interpretative guidelines (e.g., Clinical Laboratory Standards Institute [CLSI] or European Committee on Antimicrobial Susceptibility Testing [EUCAST]). Data from many pathogens and antimicrobial agents can be combined to form antibiograms, which are tables of antimicrobial susceptibility data that can be used for surveillance and to formulate population-level antimicrobial policy.

The combinations of antimicrobial agents that make up AST panels in clinical laboratories are decided using local standard operating procedures based on fixed, one-size-fits-all approaches that do not consider an individual patient's pre-test probability of resistance[5]. Many individual susceptibility tests are performed that are not reported to clinicians because they are deemed to be of limited clinical value or even potentially harmful. AST is therefore inefficient because it tests agents that produce results that are unlikely to have a positive impact on patient care, imprecise because it fails to encourage treatment that strikes the balance between individual and population benefit, and inequitable because it treats all patients the same regardless of their risk and needs.

Here, we describe how the precision, efficiency, and equitability of AST could be improved to address the global AMR problem by using a personalised approach based on clinical prediction modelling. As an exemplar, we use statistical clinical prediction approaches applied to real-world healthcare data to estimate individual probability of antimicrobial susceptibility in patients with urinary tract infection (UTI)— we use these predictions to develop a personalised AST approach that prioritises testing of WHO Access agents if their predicted probability of susceptibility is >50%. In this work, we show that this personalised approach can provide more AST results that inform the use of Access category antimicrobial agents than a standard AST approach.

## Results

### Prior AMR was the most consistently informative predictor of AMR

Key characteristics of the study population used to train and validate the clinical prediction models (model development dataset) and the population used for the individual-level simulation study (microsimulation study dataset) are summarised in Supplementary Data 1. The patient variables (and associated coefficients) used in binary logistic regression models to predict the probability of organism susceptibility to each antimicrobial agent in each urine specimen are listed in Supplementary Data 2. For 10 of the 12 antimicrobial agents assessed (all agents apart from meropenem and piperacillin/tazobactam), prior resistance to an antimicrobial agent within the last year was an informative predictor of resistance (an 'R' result) for that agent. For all 12 agents, antimicrobial treatment in the last week (in some cases with the agent in question, in other cases with a different agent) was an informative predictor of resistance. Ceftriaxone treatment within the last week or year was an informative predictor of resistance for all agents apart from ciprofloxacin and nitrofurantoin. Being in a younger age category (18–29 or 30–39) was an informative predictor of susceptibility (an 'S' result) for most antimicrobial agents. For several antimicrobial agents, a prior susceptible result for that agent within the last year was an

informative predictor of resistance to another agent and vice versa, probably reflecting recurrent growth of organisms with relatively predictable susceptibility patterns (e.g., meropenem resistance and vancomycin susceptibility in *Enterococcus faecium*).

### Susceptibility prediction performance varied depending on the drug

Figure 1 displays the receiver operating characteristic (ROC) curves and area under ROC curve (AUC-ROC) values for binary logistic regression models when they were applied to a single validation dataset for all 12 antimicrobial agents. All model validations yielded AUC-ROC values greater than 0.6 for predicting a susceptible result, and AUC-ROC values varied between antimicrobial agents. Overall, AUC-ROC values were highest for piperacillin/tazobactam (0.86) and lowest for ampicillin (0.63), with AUC-ROC values for most other antimicrobial agents falling in the 0.7–0.8 range. Table 1 summarises precision, recall, f1-score, and accuracy performance metrics for each model and for each result class on the single validation dataset, which varied between antimicrobial agents in a similar pattern to AUC-ROC values. The largest discrepancy between prediction performance for susceptible and resistant results was for piperacillin/tazobactam (f1-score 0.91versus 0.31 respectively), a phenomenon likely to be due to the rarity of piperacillin/tazobactam resistance in the dataset.

### Using less training data had minimal effect on predictive performance

A small training dataset stability analysis for all prediction models is summarised in Supplementary Figs. 1–12. When repeatedly training and testing 100 models using progressively smaller training datasets (the smallest being 2% of the model development dataset), the largest difference between the mean AUC-ROC value and the AUC-ROC value in the model described above was −0.074, for piperacillin/tazobactam at a training dataset size of 2%. The largest standard deviation of AUC-ROC values was 0.022, also for piperacillin/tazobactam at a training dataset size of 2%.

### Susceptibility prediction performance varied across demographics

Supplementary Data 3 summarises the model fairness analysis based on demographic characteristics in the dataset. Susceptibility prediction accuracy was higher in females than males for all antimicrobial agents except ampicillin. Prediction accuracy was highest in patients in the youngest age category (18–29 years) for all antimicrobial agents except ampicillin and trimethoprim/sulfamethoxazole, and usually fell with increasing age. Prediction accuracy was more variable and difficult to interpret in under-represented racial groups due to small numbers. When assessed on the microsimulation dataset, susceptibility prediction accuracy was higher for white patients (the most prevalent racial cohort in the dataset) than non-white patients in aggregate for ampicillin (0.579 vs 0.560) and lower for all other agents, with the largest difference being for meropenem (0.624 vs 0.760).

### Using more specimen information affected predictive performance

A laboratory specimen pathway simulation analysis (designed to assess how incorporating more organism information that becomes available during the specimen pathway would affect predictive performance) is summarised in Fig. 2. Including organism genus or species as a predictor variable increased AUC-ROC values for all antimicrobial agents— the greatest benefit was for meropenem (an increase in AUC-ROC from 0.728 to 0.999), and the least benefit was for ciprofloxacin (AUC-ROC increase from 0.670 to 0.741). The impact of the subsequent availability of all other AST results as predictor variables was less consistent

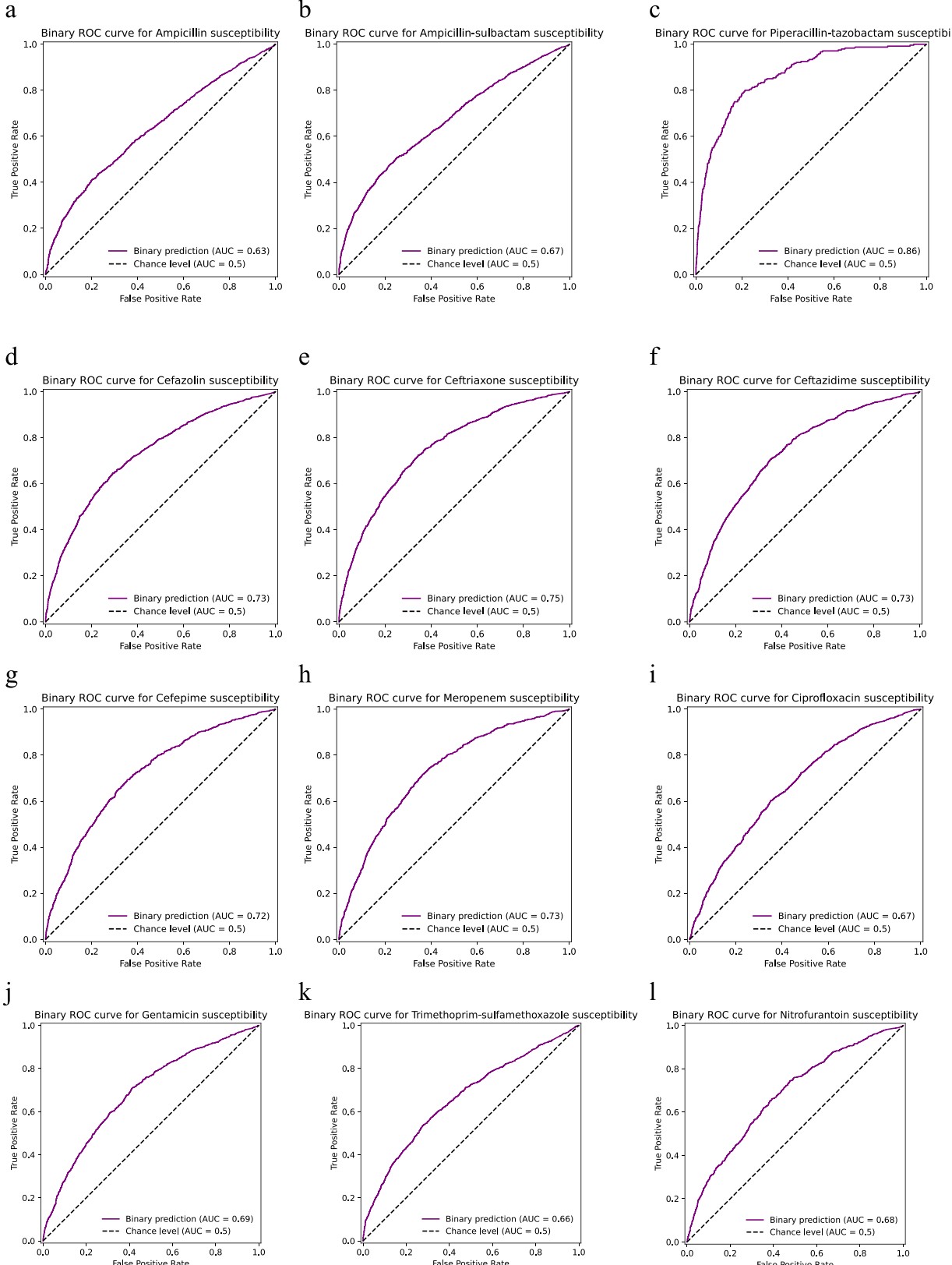

**Fig. 1 | Performance of probability predictions on the validation dataset.** Plots (**a**–**l**) correspond to the 12 antimicrobial agents for which susceptibility prediction models were developed. Purple lines represent receiver operating characteristic (ROC) curves for the predictive value of binary logistic regression models for susceptible ('S') results when applied to a validation dataset, with chance level (level of performance if the model had no predictive value) represented by black dashed lines, and area under the curve (AUC) provided in inset boxes. Source data are provided as a Source Data file.

## Table 1 | Model performance metrics

| Antimicrobial agent | Result | Precision | Recall | F1-score | Accuracy | Support |
|---|---|---|---|---|---|---|
| Ampicillin | Resistant | 0.67 | 0.44 | 0.53 | 0.59 | 2482 |
| .. | Susceptible | 0.56 | 0.76 | 0.64 | 0.59 | 2281 |
| Ampicillin/sulbactam | Resistant | 0.42 | 0.52 | 0.46 | 0.67 | 1326 |
| .. | Susceptible | 0.80 | 0.73 | 0.76 | 0.67 | 3437 |
| Piperacillin/tazobactam | Resistant | 0.19 | 0.73 | 0.31 | 0.83 | 240 |
| .. | Susceptible | 0.98 | 0.84 | 0.91 | 0.83 | 4523 |
| Cefazolin | Resistant | 0.62 | 0.63 | 0.62 | 0.68 | 1986 |
| .. | Susceptible | 0.73 | 0.72 | 0.73 | 0.68 | 2777 |
| Ceftriaxone | Resistant | 0.56 | 0.65 | 0.60 | 0.70 | 1655 |
| .. | Susceptible | 0.79 | 0.72 | 0.76 | 0.70 | 3108 |
| Ceftazidime | Resistant | 0.46 | 0.65 | 0.54 | 0.68 | 1354 |
| .. | Susceptible | 0.83 | 0.69 | 0.76 | 0.68 | 3409 |
| Cefepime | Resistant | 0.37 | 0.64 | 0.47 | 0.68 | 1061 |
| .. | Susceptible | 0.87 | 0.69 | 0.77 | 0.68 | 3702 |
| Meropenem | Resistant | 0.32 | 0.66 | 0.43 | 0.68 | 879 |
| .. | Susceptible | 0.90 | 0.68 | 0.78 | 0.68 | 3884 |
| Ciprofloxacin | Resistant | 0.33 | 0.60 | 0.43 | 0.64 | 1059 |
| .. | Susceptible | 0.85 | 0.65 | 0.73 | 0.64 | 3704 |
| Gentamicin | Resistant | 0.36 | 0.61 | 0.45 | 0.65 | 1124 |
| .. | Susceptible | 0.85 | 0.67 | 0.75 | 0.65 | 3639 |
| Trimethoprim/sulfamethoxazole | Resistant | 0.55 | 0.55 | 0.55 | 0.64 | 1896 |
| .. | Susceptible | 0.70 | 0.70 | 0.70 | 0.64 | 2867 |
| Nitrofurantoin | Resistant | 0.28 | 0.57 | 0.37 | 0.66 | 856 |
| .. | Susceptible | 0.88 | 0.68 | 0.77 | 0.66 | 3907 |

Additional performance metrics for the 12 clinical prediction models when applied to the single validation dataset.

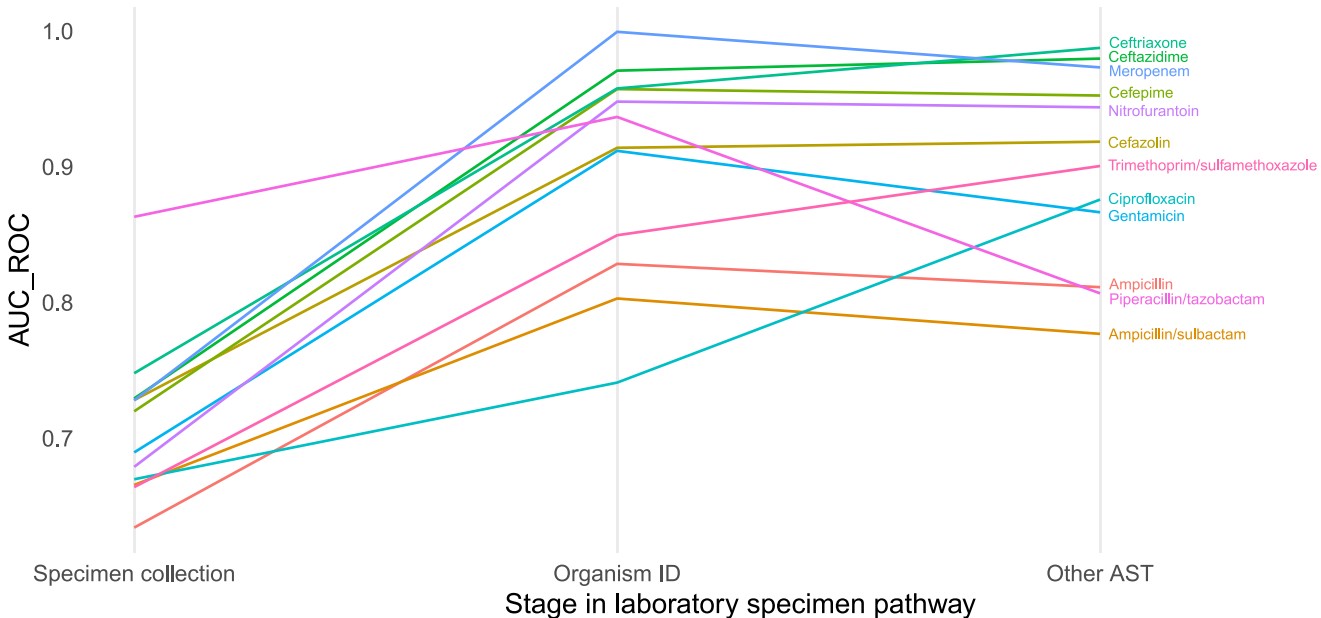

**Fig. 2 | Specimen pathway sensitivity analysis.** Susceptibility prediction performance throughout the simulated laboratory specimen pathway expressed as AUC-ROC values, as further information becomes available in the form of organism ID (identification) and other antimicrobial susceptibility testing (AST) results. Source data are provided as a Source Data file.

—the largest benefit was for ciprofloxacin (AUC-ROC increase from 0.741 to 0.876), but AUC-ROC values fell for several agents, the most marked example being piperacillin/tazobactam (AUC-ROC decrease from 0.937 to 0.807).

**Multinomial models did not improve predictive performance**
The main analysis was performed using binary logistic regression, defining susceptible as a positive result. For most antimicrobial agents, repeating the analysis using multinomial logistic regression to predict

**Fig. 3 | Microsimulation study results-per-specimen analysis.** The number of susceptible results per specimen provided by application of the personalised approach (PDAST) and standard fixed approach in the individual-patient simulation (microsimulation) study for all agents (left) and for WHO Access agents (right). The six results were chosen based on the PDAST or standard panel recommendations from the 12 actual real-world results that were available in the study dataset, representing a simulation of a real-world testing decision and outcome. Blue and red circles represent median values, blue and red lines represent the interquartile range, and grey dot cloud darkness represents the number of results from all specimens. Source data are provided as a Source Data file.

susceptible, intermediate, resistant, and non-testable results separately (Supplementary Fig. 13) returned lower AUC-ROC values on average than the main analysis (depending on the averaging strategy used). Lower AUC-ROC values for relatively rare intermediate ('I') results brought average AUC-ROC values down—use of a class frequency-weighted 'micro-averaging' approach therefore increased average AUC-ROC values.

### Out-of-sample predictive performance varied minimally over time

When binary logistic regression models were repeatedly trained on data from one time period and validated on data from another using four time windows (2008–2010, 2011–2013, 2014–2016, and 2017–2019), the largest difference in mean AUC-ROC between validations was 0.059, observed when the ciprofloxacin model trained on data from 2011–2013 was validated on data from 2008–2010 (AUC-ROC 0.666) and 2017–2019 (AUC-ROC 0.607). The largest standard deviation of AUC-ROC values across a 20-run cross-validation was 0.034, observed for piperacillin/tazobactam using training data from 2017–2019 and validation data from 2014–2016. The distributions of AUC-ROC values for the out-of-sample analysis are displayed in Supplementary Figs. 14–25.

### Personalised AST provided more susceptible WHO Access results

Figure 3 summarises the results of the individual-level simulation (microsimulation study) that measured the number of susceptible results that would be provided per specimen by our personalised approach (in which WHO Access agents were automatically tested if their predicted probability of susceptibility was >50%, then the rest of the six-agent panel was filled in descending order of probability of susceptibility), and the standard fixed-panel approach (testing piperacillin/tazobactam, nitrofurantoin, ciprofloxacin, gentamicin, ceftriaxone, and trimethoprim/sulfamethoxazole on all specimens). On average, the personalised AST approach provided more susceptible results per specimen for WHO Access agents than the standard approach (effect size 0.32; $P < 0.001$) but provided fewer susceptible results per specimen in general (effect size −0.19; $P < 0.001$). The personalised AST approach provided at least one susceptible Access agent result less frequently than the standard panel (76.9% vs 96.6% of specimens; $P < 0.001$; 95% CI of difference −21.9% to −17.9%) but provided at least one susceptible result of any kind in a similar proportion of cases (98.2% vs 98.9% of specimens; $P = 0.066$; 95% CI of difference −1.3% to 0.0%).

### Lowering decision thresholds improved personalised AST performance

Figure 4 summarises the results of an analysis that examined the effects of changing the decision threshold for testing Access agents. Automatically testing WHO Access agents with >30% probability of susceptibility (a lower threshold than the default 50% used in the main analysis) increased the median number of susceptible Access results per specimen from 3 to 4 while still providing a median 4 susceptible

a

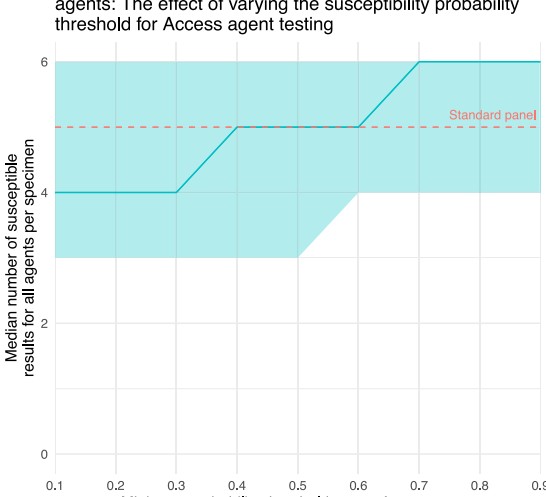

b

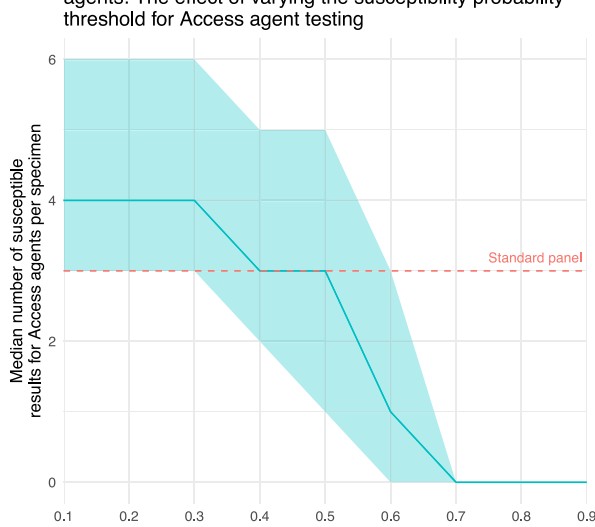

c

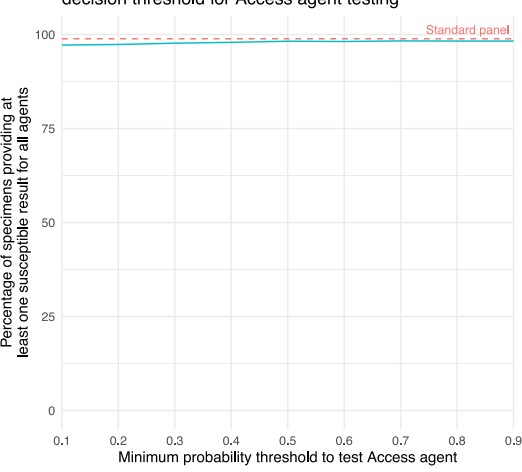

d

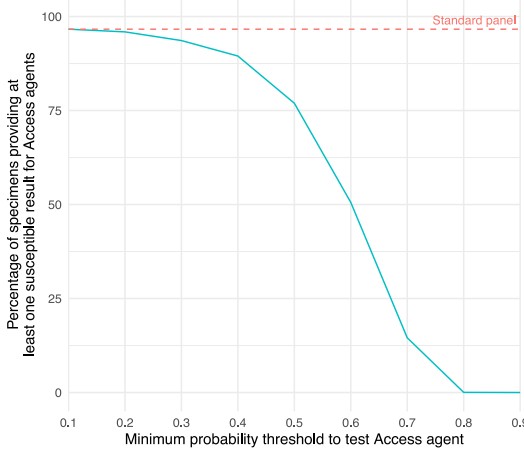

**Fig. 4 | Microsimulation study decision threshold sensitivity analysis.**
**a**, **b** display the effect of varying testing decision threshold on number of susceptible results per panel for all agents and WHO Access category agents respectively. Plots c and d display the effect of varying testing decision threshold on the percentage of panels with least one susceptible result for all agents and WHO Access category agents respectively. In plots a and b, blue lines represent median number results per specimen overall, and increased the proportion of specimens providing at least one susceptible Access result from 76.9% to 93.6% while still providing a susceptible result of any kind for 97.7% of specimens. When the analysis was repeated with all 'I' results reclassified as resistant, the personalised approach still provided more susceptible Access results on average than the standard approach, which was more sensitive to the reclassification–the median number of susceptible Access results provided by the standard approach at the

of susceptible results per panel for the personalised approach (PDAST), shading represents interquartile range. In plots (**c**, **d**) blue lines represent percentage of specimens providing at least one susceptible result. Red dashed lines represent median number of susceptible results per specimen (**a**, **b**) and percentage of specimens with at least one susceptible result (**c**, **d**) with the standard panel. Source data are provided as a Source Data file.

50% decision threshold fell from 3 to 2. The results of this analysis are summarised in Supplementary Figs. 26 and 27.

**Personalised AST detected opportunities to use Access beta-lactams**
Figure 5 summarises the total number of susceptible results available for each antimicrobial agent using the personalised and standard AST approaches. The largest differences in antimicrobial agent

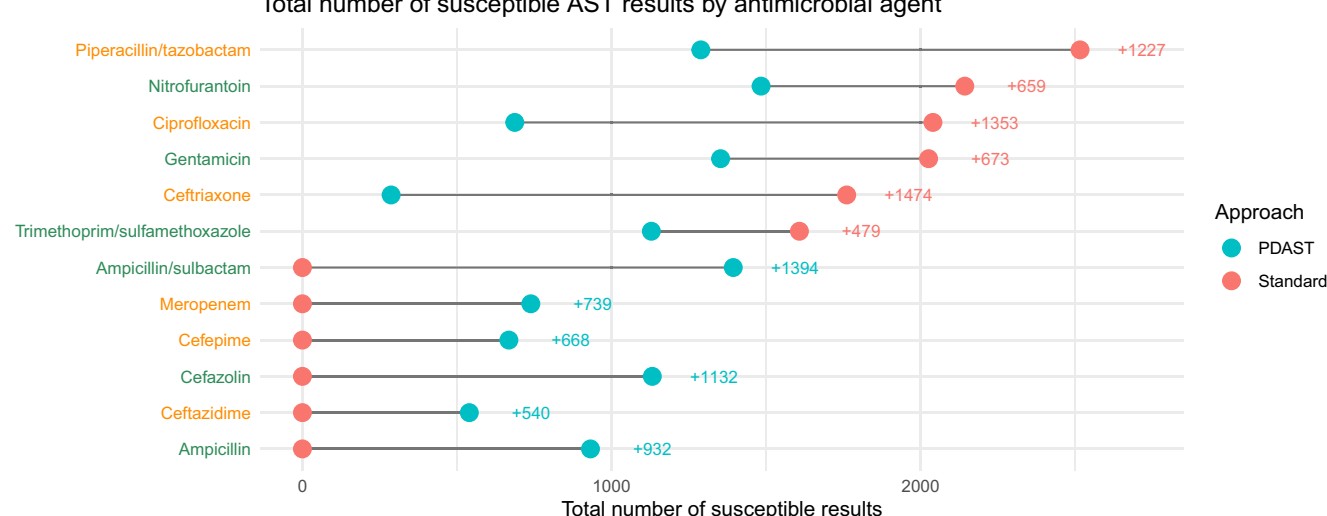

**Fig. 5 | Microsimulation study results-per-antimicrobial agent analysis.** The total number of susceptible antimicrobial susceptibility testing (AST) results provided by the personalised (PDAST) approach and standard approach for all 12 antimicrobial agents studied. WHO Access category agent axis labels are green in colour, while those for Watch category agents are orange. Source data are provided as a Source Data file.

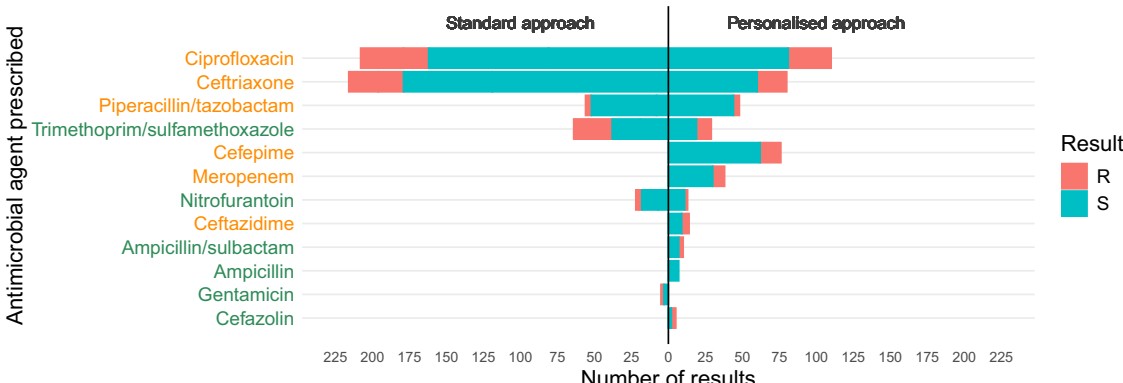

**Fig. 6 | Results provided for the antimicrobial agent inpatients were prescribed.** Bar size corresponds to the number of prescriptions, with susceptible (S) results in green and resistant (R) results in red. Counts for the standard approach are displayed to the left of the central vertical line, and counts for the personalised approach are displayed to the right of the line. Source data are provided as a Source Data file.

testing frequency between the personalised and standard approaches were for the Access agents ampicillin, ampicillin/sulbactam, and cefazolin (tested more frequently in the personalised approach), and the Watch agents piperacillin/tazobactam, ceftriaxone, and ciprofloxacin (tested less frequently in the personalised approach).

### Personalised AST provided fewer results for prescribed agents

The personalised approach provided 430 results for antimicrobial agents that inpatients were currently being administered, versus 571 results provided by the standard panel —this difference was mainly due to prescriptions for ceftriaxone and ciprofloxacin, the most commonly-prescribed agents in patients with growth in urine specimens (see Fig. 6). The personalised approach provided 46 susceptible results for Access category agents that patients were currently being administered, versus 59 comparable results provided by the standard panel—this difference was mainly due to prescriptions for trimethoprim/sulfamethoxazole and nitrofurantoin.

### Personalised AST did not facilitate more Access oral step-downs

The personalised approach provided at least one susceptible Access category result for patients prescribed a Watch category agent in 386 instances, compared to 618 instances using the standard approach—this difference was mainly due to the presence of gentamicin on the standard panel, an Access agent with a relatively low rate of resistance in this dataset (see Supplementary Data 1). The personalised approach provided susceptible results for Access category agents with oral routes of administration (ampicillin, ampicillin/sulbactam, trimethoprim/sulfamethoxazole, and nitrofurantoin) for patients on parenteral-only agents (ceftriaxone, piperacillin/tazobactam, cefepime, meropenem, ceftazidime, gentamicin, and cefazolin) with a similar frequency to the standard approach (648 cases versus 669).

### Discussion

Our personalised AST approach could inform antimicrobial prescribing to help achieve the United Nations General Assembly target of at least 70% WHO Access agent use globally by 2030. Choosing an appropriate susceptibility probability threshold at which Access

agents are automatically tested could achieve this goal without endangering the provision of other treatment options. Furthermore, it could help clinicians and health systems achieve the core aim of antimicrobial treatment—effectively treating the individual while minimising harm to that individual and the wider population.

When embedded in routine microbiology laboratory workflows (see Supplementary Fig. 28), our personalised AST approach could be used effectively in a range of resource settings, because it can potentially provide more useful results without expending resource on testing more antimicrobial agents per specimen. It could also be combined with, or embedded in, existing solutions such as WHONET laboratory database software[6]. The potential for our approach to provide more Access category treatment options may also benefit patients in low and middle-income countries that lack state-funded healthcare because Access agents are typically cheaper than Watch agents[7].

There are two key elements to developing personalised AST and other adaptive data-driven microbiology diagnostics: firstly, predicting the result; and secondly, prioritising tests based on the potential impact of their results on patient and population outcomes. For predicting AST results, logistic regression is a suitable method for several reasons: firstly, it requires relatively little computational power (all algorithms were developed and run on a laptop computer running macOS Sonoma version 14.5 with an Apple M1 Pro processor, 16GB random-access memory and 10 cores) and can therefore be deployed in a range of global resource settings; secondly, its cost function (log-loss) is convex, meaning it is easier to optimise model parameters using the relatively simple technique of gradient descent; and thirdly, clinicians can understand how predictions were made by interrogating the model coefficients (see Supplementary Data 2)[8].

The models used in this study had a varying degree of predictive performance across the 12 antimicrobial agents tested—there are several potential reasons for this: firstly, agents such as ampicillin (or amoxicillin) are predominantly administered in the community—unobserved, unmeasurable variables (i.e., events that happen outside of hospital) may therefore have more impact than the available variables derived from electronic healthcare records[9]; secondly, the type, number, complexity, and phenotypic expression of resistance mechanisms that contribute to AMR vary from agent to agent and organism to organism (for example, carbapenem resistance can occur via several mechanisms, each of which is linked to different patient risk factors)[10]; thirdly, for agents with class imbalance (i.e., very high or low rates of resistance in the dataset), a small number of correct or incorrect predictions can result in disproportionate changes in AUC-ROC values[11].

Applying arbitrary AUC-ROC performance thresholds and other performance metrics to determine so-called good or bad predictions is not meaningful for clinical implementation of the susceptibility prediction model for personalised AST, in which event the consequences of predictions are decision-, antimicrobial-, patient-, and context-specific[12]. It is for this reason that we performed an individual patient-level microsimulation study to evaluate the potential clinical impact of personalised AST. It is also important to understand where in the specimen pathway predictions need to be made, because this influences the strength and usefulness of susceptibility prediction (see Fig. 2). For example, piperacillin/tazobactam susceptibility predictions would not need to be made for intrinsically resistant organisms (e.g., *E. faecium*) or where susceptibility can be inferred (e.g., ampicillin/sulbactam-susceptible Enterobacterales)—once these cohorts are removed by learning organism identification and AST results, susceptibility prediction performance may be poorer in the remaining selected cohort (e.g., ampicillin/sulbactam-resistant Enterobacterales). Adapting standard operating procedures and laboratory information management systems alongside personalised AST could improve predictive performance early in the specimen pathway, e.g.,

by integrating additional healthcare data from primary care, or implementing rapid organism identification methods.

Understanding the patient pathway is also crucial to implementing personalised AST in complex healthcare systems. The approach will need to strike the balance between the two main Access agent outcomes measured in this study—i.e., between providing more susceptible Access results per specimen (an approach that may be useful where there are drug contraindications, e.g., nitrofurantoin in patients with renal impairment, or penicillin allergy) and providing at least one susceptible Access result per specimen (an approach that hinges on the usefulness of that antimicrobial agent in that clinical context, e.g., whether it is an orally-administrable agent for a patient in general practice). Understanding local prescribing practice and the prevalence of AMR is also important. For example, personalised AST probably delivered fewer results for prescribed agents in this study because the most likely local antimicrobial formulary recommendations (ceftriaxone and ciprofloxacin) were both Watch category agents, and it probably failed to provide an oral step-down benefit due to low rates of nitrofurantoin and trimethoprim-sulfamethoxazole resistance (oral Access agents both included on the standard panel). In real-world practice, our probabilistic approach is flexible enough for these issues to be mitigated by combining personalised AST with fixed standard operating procedures—for example, applying rules to always test screening agents for extended-spectrum beta-lactamases / AmpC enzymes, ensuring an oral agent is always tested, or to prioritise/protect agents often used in an organism-specific context (e.g., ceftazidime for *Pseudomonas aeruginosa*). Improved real-time healthcare data linkage would also allow for automatic testing of the agent the patient had been prescribed if this was desirable.

Practical constraints will also need to be considered to implement personalised AST in real-world settings—for example, our personalised approach could be slow and cumbersome if using manual AST disc dispensers. Our prototype application (Supplementary Fig. 28) can therefore recommend the best overall panels for a single laboratory testing session (so-called session panels), allowing a user-determined number of disc dispensers to be pre-loaded for a testing session. The application can also re-order specimens to minimise the number of differences between consecutive panels to minimise change frequency (so-called efficiency-optimised ordering), in case only one disc applicator was available. Further work could assess the ability of these practical solutions to provide Access antimicrobial treatment options in simulated and real-world settings. A process of co-design with practitioners and patients will be required to ensure personalised AST is a practical and feasible model for a range of global healthcare environments. For real-world implementation to then proceed, sufficient resource and infrastructure would need to be provided to monitor and maintain performance of the clinical prediction model and the intervention. Furthermore, the legislative and regulatory requirements for implementing the clinical prediction model in the clinical laboratory setting will need to be clearly understood.

Our study has several limitations: firstly, although AST results were available for 12 antimicrobial agents, missing data prevented the full range of commonly-used agents being tested—we cannot be certain that the approach will perform equally well for all antimicrobials used in clinical care, including those in the WHO Reserve category; secondly, the AWaRe classification was used to inform decisions and outcomes in this study because it is the most widely used method of quantifying the overall impact of antimicrobial treatment on AMR at population level, but there are a range of factors that influence the choice and individual and population outcomes of antimicrobial therapy that cannot be measured with the available data—a robust measure of the impact of personalised AST (or any intervention targeted at antimicrobial prescribing) will require a combination of mixed qualitative/quantitative research techniques to predict clinician

behaviour in response to AST results, and targeted data collection (e.g., population and wastewater screening) combined with laboratory technological advances (e.g., microbiomics) to better quantify overall population impact; thirdly, the prediction models developed in this single-area study population performed better in common UTI cohorts with relatively little risk of AMR, i.e., females and younger people—there was also variation amongst racial groups that needs to be better-understood. Further studies could incorporate algorithmic fairness techniques (e.g., threshold optimisation), but the most effective solution will be to train and validate models across more diverse patient cohorts in multiple global settings to ensure no patient or practitioner groups will be disadvantaged by use of personalised AST in UTI—this approach will also allow the performance of the approach in settings of different AMR prevalence and low- and middle- income countries to be assessed.

Despite these potential limitations, our results suggest that microbiology laboratories could use personalised AST as a simple, resource-efficient way of addressing the global AMR problem by facilitating the use of WHO Access category agents for UTI, without endangering the ability of AST to provide treatment options for clinicians. Furthermore, personalised AST is a template for a new

generation of adaptive real-time diagnostics that will become essential for regional, national, and global AMR preparedness. Deploying personalised AST in real-world practice will require a process of co-production with practitioners and patients, robust regulatory frameworks, more diverse training dataset patient cohorts, and suitable healthcare data science infrastructure to ensure its efficacy, equitability, and safety.

## Methods

### Setting

The study complied with the ethical regulations stipulated by the PhysioNet MIMIC (Medical Information Mart for Intensive Care)-IV Data Use Agreement and Health Data License—MIMIC-IV was accessed and used according to the terms of the PhysioNet Credentialed Health Data Use Agreement 1.5.0. The personalised approach was developed and assessed using MIMIC-IV version 2.2, an open-source, pseudonymised electronic healthcare record dataset from Beth Israel Deaconess Medical Center (BIDMC) in Boston, MA[13–15]. The dataset contains hospital-wide inpatient and outpatient data for patients who were admitted to intensive care or the emergency department between 2008 and 2019.

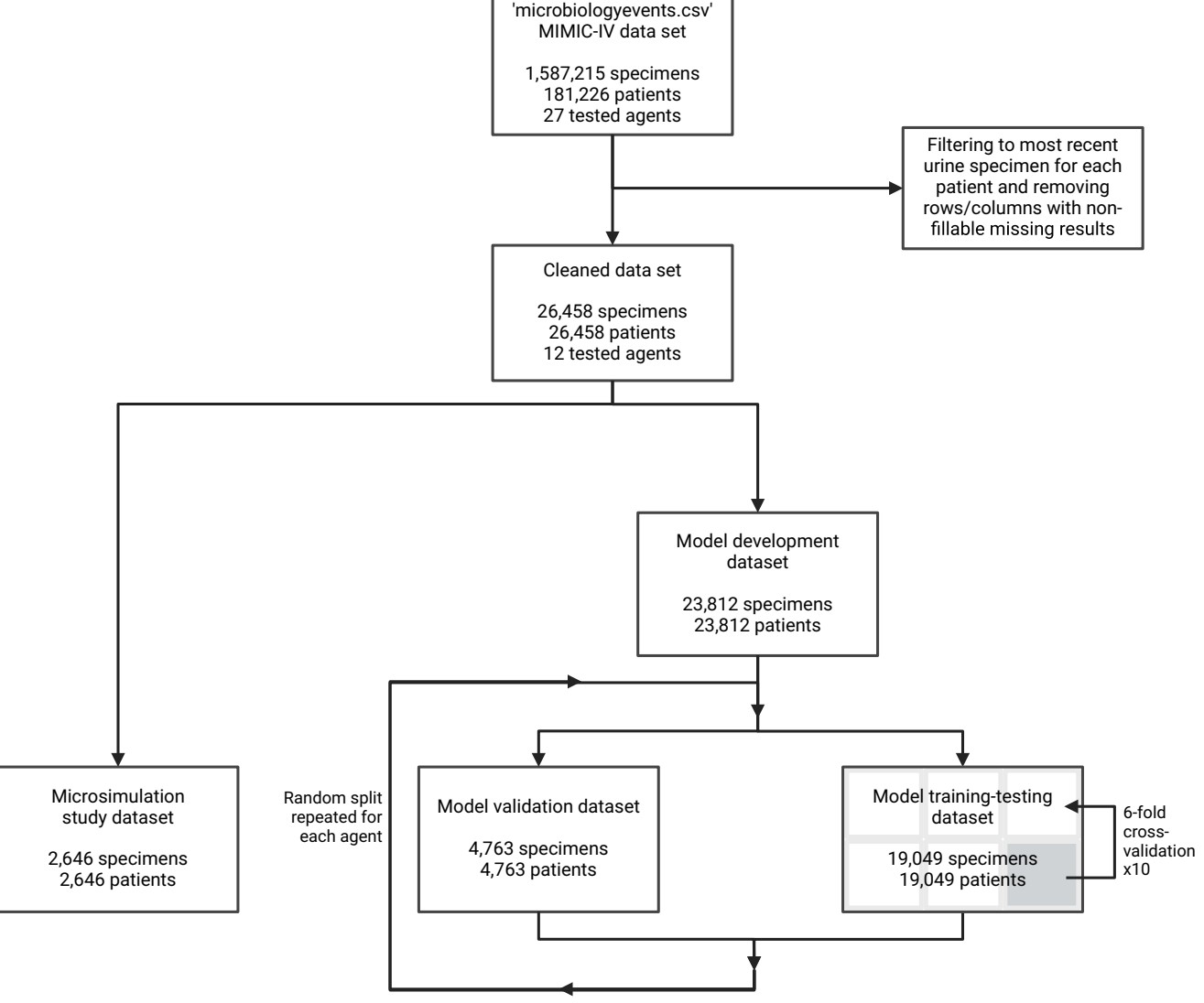

**Fig. 7 | Flow chart for the study population.** The initial dataset was cleaned, then split into model development and microsimulation datasets. The model development dataset was subsequently split into training-testing dataset used for hyperparameter tuning and feature selection, and a validation dataset used once to measure predictive performance. Created in BioRender. Howard, A. (2024) BioRender.com/z80x717.

## Participants and study size

No statistical method was used to predetermine sample size, which was instead determined by the size of the available dataset. Figure 7 displays the flow chart for the study population and Supplementary Table 1 summarises the data cleaning process, including the management of missing data—this process yielded a final dataset with complete AST results for 13 agents (ampicillin, ampicillin/sulbactam, piperacillin/tazobactam, cefazolin, ceftriaxone, ceftazidime, cefepime, meropenem, ciprofloxacin, gentamicin, trimethoprim/sulfamethoxazole, nitrofurantoin, and vancomycin) on 26,458 urine specimens for 26,458 patients. The full analysis was not pursued for vancomycin because of the rarity of its use in UTI, leaving 12 agents with complete AST results for the study. Only the most recent urine specimen was used for each patient to prevent cross-pollution of training, validation, and microsimulation datasets by multiple specimens from the same patient(s). The dataset was randomly split by sampling without replacement at specimen-level into a model development (training and validation) dataset (90% of specimens) and a microsimulation study dataset (10% of specimens). The Investigators were not blinded to allocation during experiments and outcome assessment.

## Data sources/measurement and quantitative variables

Available classes of antimicrobial resistance results were susceptible (S), intermediate (I), and resistant (R). These interpretations are provided by the MIMIC-IV dataset—the interpretative criteria are not cited but are likely to be CLSI for US-derived data and are likely to be based on MIC data that are also provided. An additional class, NT (not testable) was added for specific organism-drug combinations where susceptibility of the organism is unclear/not measured (e.g., meropenem in some *Enterococcus* spp.). For the binary probability predictions made in the main analysis, susceptible (S) or intermediate (I) results were classified as susceptible (on the assumption that concentrations of antimicrobial agents achieved in urine are likely to achieve the exposure required for organisms with I results[16]), while resistant (R) and not testable (NT) results were classified as resistant. Organism data were available to species level for all common organisms—the only exception to this is *Enterococcus* spp., for which only genus-level data were usually available. Predictor variables eligible for inclusion in models were healthcare data from the datasets as summarised in Supplementary Data 2, chosen using a logical approach based on plausible causal interactions between feature variables and antimicrobial resistance[17].

## Descriptive analyses and clinical prediction model development

Data cleaning and statistical analyses of model stability and outcomes were performed in R version 4.3.2 (https://cran.r-project.org) using AMR package version 2.1.1[18]. Splitting multiclass variables into Boolean variables, mathematical modelling, and statistical analyses of model variables and performance metrics were performed using Python version 3.12.0 (https://www.python.org/downloads/release/python-3120/)[20–21].

A separate model was trained, tested, and validated on a model development dataset for each of the 12 antimicrobial agents. The workflow performed for each antimicrobial agent was as follows:

1. The model development dataset was randomly split into a training and hyperparameter optimisation dataset (80% of the model development dataset) and a validation dataset (20% of the model development dataset), stratified to preserve similar outcome proportions in training and validation datasets.
2. Model fitting, feature selection and hyperparameter optimisation were performed using ten sets of six-fold cross-validations with random dataset shuffling and splitting.
3. Model validation was performed once for each of the 12 antimicrobial agents using the final selected model.

## Clinical prediction model statistics and reproducibility

A separate binary logistic regression model was trained, tested, and validated for each of the 12 antimicrobial agents using the Scikit-Learn package version 1.4.1 (https://scikit-learn.org/stable/) and Mlxtend version 0.23.1[22,23]. Susceptible results were defined as positives, and resistant results were defined as negatives. Model training was conducted using maximum likelihood estimation. Trained models were used to output probabilistic predictions for each specimen. A least absolute shrinkage and selection operator (LASSO, or L1) regularisation technique was used to prevent overfitting and eliminate predictor variables that were found to have little influence on predictions[24]. The strength of regularisation was determined by hyperparameter tuning, performed using grid search cross-validation, to optimise AUC-ROC. AUC-ROC values for susceptibility prediction were calculated using the scikit-learn roc_auc_score function. Precision, recall, accuracy, and f1-scores (using a decision threshold of 50%) were calculated using the scikit-learn classification_report method to assess the performance of each logistic regression model on the validation dataset, with all values apart from accuracy reported for both susceptible and resistant classes.

An out-of-sample sensitivity analysis was performed by splitting the model development dataset into four 3-year time periods (2008–2010, 2011–2013, 2014–2016, and 2017–2019). The binary logistic regression technique described above was then cross-validated using 20 random 80:20% train-test splits across every possible combination of time periods, for each of the 12 antimicrobial agents (3840 train-validation runs in total). For example, training data from 2008–2010 would be validated on the remaining 20% of 2008–2010, then 20% of 2011–2013, then 20% of each of the remaining two time periods. AUC-ROC values were calculated for each of these runs and plotted to compare their distributions in sample and out of sample. Mean and standard deviations of AUC-ROC values for each combination of time periods were calculated.

A stability analysis was performed on 100 different random splits in the model development dataset at each of a range of smaller training-testing dataset split ratios (16:84%, 14:86%, 12:88%, 10:90%, 8:92%, 6:94%, 4:96%, and 2:98%)—a visual analysis of the mean and distribution of probability predictions, AUC-ROC values, and mean absolute prediction error was then undertaken across the range of train-test dataset split ratios. A model fairness analysis was performed, in which the accuracy, true positive rate, false positive rate, false negative rate, and predicted-as-positive rate (using a decision threshold of 50%) were calculated for each demographic group based on the variables 'gender', 'age', 'race', 'language', 'insurance', and 'marital status'. A separate analysis was then performed where 'susceptible', 'intermediate', 'resistant', and 'not testable' were treated as separate classes and a one-vs-rest multinomial logistic regression approach was used, then micro-and macro-averaged AUC-ROC values across the four classes were calculated.

The main analysis assumed that predictions would need to be made at the beginning of the laboratory specimen pathway before any results from the specimen itself were available. A laboratory specimen pathway simulation analysis was therefore performed to examine the effects of including organism identification and AST information that may be obtained later in the specimen pathway. Firstly, organism identification was included as a predictor in the model, and AUC-ROC values were recalculated for all antimicrobial agents. *Enterococcus* species were excluded from this analysis (on account of having less than six agents in the 12 with intrinsic activity, meaning a personalised AST approach would have little value once the organism is identified), as were organisms with intrinsic resistance (for example, ampicillin predictions were not made for *K. pneumoniae*). The process was then repeated with additional inclusion of other AST results as predictor variables. Specimens in which

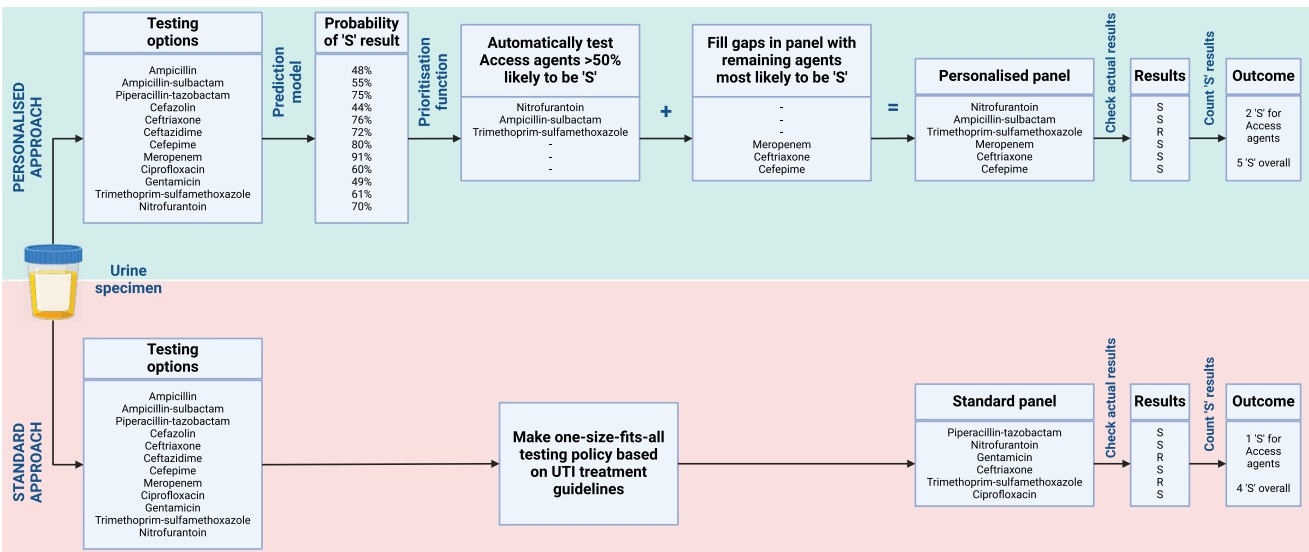

**Fig. 8 | Design of the microsimulation study.** For each specimen in the micro-simulation study dataset, a personalised panel was composed using prediction modelling and a test prioritisation function to maximise the probability of susceptible ('S') results for Access agents, and failing that, susceptible results for other agents—the results for agents in this panel were then populated by actual results for those agents in the dataset. The number of susceptible results for Access agents and all agents that would have been provided by using this panel was then compared against a standard panel based on international UTI treatment guidelines. Created in BioRender. Howard, A. (2024) BioRender.com/k67g601.

available AST results had deterministic relationships with the dependent variable (e.g., ampicillin/sulbactam resistance when predicting ampicillin susceptibility) were additionally excluded from this part of the analysis. AUC-ROC values that could be achieved by the clinical prediction model at the three different points in the laboratory specimen pathway were then analysed visually.

**Microsimulation study design**
Figure 8 summarises the study design for the microsimulation (individual patient-level simulation) study.

The microsimulation study was performed by interfacing R version 4.3.2 and Python version 3.12.0 using the R package reticulate version 1.36.0 (https://cran.r-project.org/web/packages/reticulate/index.html)[25]. For each specimen, six of the 12 agents were selected as the simulated personalised testing panel, by prioritising based on probability of susceptibility—WHO Access agents with >50% probability of susceptibility were tested automatically, then the rest of the six-agent testing panel was filled with remaining agents that had the highest probabilities of susceptibility (regardless of whether they were Access or Watch category agents). The personalised panel for each specimen was then compared with a standard non-personalised testing panel composed of four antimicrobial agents recommended for UTI by the WHO essential medicines list (nitrofurantoin, trimethoprim/sulfamethoxazole, ceftriaxone, and ciprofloxacin) and two agents recommended by European Association of Urology guidelines (gentamicin and piperacillin/tazobactam)[26,27].

**Outcome variables**
The outcomes of interest were:
- The number of susceptible results per specimen six-agent panel.
- The number of susceptible results for Access agents per specimen six-agent panel.
- The proportion of specimen six-agent panels with at least one susceptible result.
- The proportion of specimen six-agent panels with at least one susceptible result for an Access agent.

**Microsimulation study statistics and reproducibility**
Numbers of susceptible results per panel were compared between the personalised and standard fixed panel on each specimen using a Wilcoxon two-sample paired signed rank test with continuity correction—results are presented as effect size (calculated by dividing the test's Z statistic by the square root of the number of specimens)[28]. Medians and interquartile ranges of the number of susceptible results per panel were calculated for data visualisation purposes. Proportions of specimens with panels with at least one susceptible result were compared using a chi-squared test—results are reported as proportions for the two groups and 95% confidence interval. For both analyses, a significance threshold of 5% was used. A sensitivity analysis was also performed for the test prioritisation function, where the analysis was repeated across nine different susceptibility probability thresholds for automatic inclusion of Access agents on the personalised panel (from >10% probability of susceptibility to >90%). This sensitivity analysis was then repeated after also repeating the modelling and panel selection steps, but with all intermediate (I) results classified as resistant instead of susceptible.

**Reporting summary**
Further information on research design is available in the Nature Portfolio Reporting Summary linked to this article.

# Data availability
The MIMIC-IV version 2.2 raw data are available under restricted access for ethical reasons, access can be obtained as a credentialed PhysioNet user at https://physionet.org/content/mimiciv/2.2/ once mandated training is completed and the data use agreement is signed. The aggregate data generated in this study are provided in the Supplementary Information and Supplementary Data files. Source data are provided with this paper.

# Code availability
The results of the study can be reproduced in full using the above open-source data and open-source code available at https://github.com/amh312/PDAST/ (https://doi.org/10.5281/zenodo.13920515). If

required, an abbreviated version of the analysis can be run on a small, simulated dataset at CodeOcean (https://doi.org/10.24433/CO.7794737.v1).

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

## Acknowledgements

This research was funded in part by the Wellcome Trust [grant ref: 226691/Z/22/Z] (WH). For the purpose of open access, the author has applied a CC BY public copyright licence to any Author Accepted Manuscript version arising from this submission. Office for Life Sciences Data-Action Accelerator award also supported this work. The funders had no role in the conceptualisation, design, data collection, analysis, decision to publish or preparation of the manuscript. We thank Dr Matthew Sperrin and Professor Mike Sharland for their expert input and advice.

## Author contributions

A.H. conceived the study, performed data engineering and mathematical modelling, and wrote the manuscript including diagrams. A.V. and A.G. performed data engineering. P.L.G. and D.M.H. provided editing suggestions and statistical expertise. S.M. provided statistical expertise. I.B. and W.H. provided supervision and editing suggestions.

## Competing interests

Alex Howard declares personal consulting work for Pfizer outside the submitted work, and a donation from Pfizer to the University of Liverpool for a public and professional engagement project outside the submitted work. Iain Buchan declares consulting fees via University of Liverpool from AstraZeneca outside the submitted work. William Hope holds or has recently held research grants with UKRI, EU (FP7, IMI-1, IMI-2), Wellcome, F2G, Spero Therapeutics, Antabio, Pfizer, Allecra, Bugworks, Phico Therapeutics, BioVersys, Global Antimicrobial Research and Development Partnership (GARDP). He is (or has recently been) a consultant for Appili Therapeutics, F2G, Spero Therapeutics, Pfizer, GSK, Phico Therapeutics, Pulmocide, and Mundipharma Research Ltd. He was a member of the Specialist Advisory Committee for GARDP (2020-2023), a member of the British Society for Antimicrobial Chemotherapy (BSAC) Breakpoint Committee (2020-2023), a member of Health Technology Appraisal (HTA) Prioritisation Committee for hospital care and was the Specialty National Co-lead for Infection for the National Institute of Health Research (NIHR) (2020-2024). The other authors declare no competing interests.
