## [Peer Review File · Nature Communications]

Personalised antimicrobial susceptibility testing with clinical prediction modelling informs appropriate antibiotic use

Corresponding Author: Dr Alex Howard

Version 0:

Reviewer comments:

Reviewer #1

(Remarks to the Author)

Fascinating study that applies a model to predict susceptibility results to WHO ACCESS drugs to promote their testing and reporting with the intention of increasing their use.

Very well written and clear manuscript. Do not have many specific edits, however would request the authors to comment on the following in the discussion in more clarity.

1) when choosing to apply the predication model at the start of the laboratory process, it seems to fail to select agents that despite being on the ACCESS list would be more clinically suitable for the organism or rather less suitable to use as it would be considered a organism specific drug eg. ceftazidime should be reserved Pseudomonas or meropenem would be reported when it should be reserved as 2nd or even 3rd line option. This could have the unintended consequences of driving AMR and losing these last line agents instead of driving ABX use to narrower spectrum options specific to the organism

2) the desire to report a longer list of susceptible antibiotics is not always the objective. this can inadvertently direct clinicians to choose agents that again are 2nd or third line agents.

3) It seems that it would be more helpful to help lab workflow and patient care to apply the prediction model at the time of organism identification. However, based on your findings the prediction outcome was less beneficial. Can you clarify what could be done to improve the prediction at this point in the laboratory workflow?

Reviewer #2

(Remarks to the Author)

In this manuscript the authors used MIMICv4 EMR data from the US to derive a dataset containing the contextual data and antimicrobial susceptibilities of 26,458 urine specimens. They then trained L1 or LASSO-regularised logistic regression models to predict susceptibility to 12 antimicrobials (and/or associated inhibitors). This followed best practices including cross-validated feature selection/hyperparameter tuning and a held-out validation set only used once. They also performed ancillary sensitivity and stability analyses including evaluating the requirement of data-set size, a multinomial reframing, decision threshold, and potential algorithmic biases against specific demographic groups. Following this, they used a second held out dataset to attempt to evaluate the impact of employing these models within the typical clinical microbiology lab workflow.

Finally, they determined that this predictive approach could reduce the number of AST tests being performed while increasing the number of susceptible results.

This paper, although relatively simple analytically, is technically well executed and has the potential to inform current and future work attempting to improve efficiency and outcomes by predictive personalisation of clinical microbiology lab assays. However, the results obtained are likely to of interest to largely clinical lab microbiologists/scientists rather than a broader audience. For this reason it may have greater impact in a more specialised clinical/medical microbiology journal.

Minor

Line 70 (Introduction) - given the importance of appropriate use of interpretive criteria for MIC/ZOI data this should probably at least mention them

Line 76 (Introduction) - which results are actually reported out may also vary

Line 425 (Methods) - SIR under which set of guidelines? Presumably CLSI for US-derived data?

Line 437 (Methods) - Please provide citations and not just URL links to the key libraries used - these citations are an important metric that funds their on-going development! This is specifically outlined and requested by most of the libraries used in this paper e.g., <https://scikit-learn.org/stable/about.html#citing-scikit-learn> <https://rasbt.github.io/mlxtend/> <https://rstudio.github.io/reticulate/authors.html#citation> <https://numpy.org/citing-numpy/> etc

Line 487 (Methods) - Should these details not come before the microsimulation study?

Table 1 (Results) - Is the remaining % female or are there rows with no data?

Table 2 (Results) - What was the variances of these metrics across CV folds?

Table 2 (Results) - How are metrics for both R and S reported? It is a bit unusual to report both for binary classification as it involves treating S as both a true positive and true negative in each calculation. Did you just flip the confusion matrix diagonals and recalculate? I can see the justification of providing more context but worth expanding upon a little bit due to its unusual nature.

Discussion - Another major consideration of adoption is the logistics of monitoring model performance over time and how retraining will be performed. Predictive models are generally legislated as medical device leading to considerable administrative overheads and challenges.

Reviewer #3

(Remarks to the Author)

The paper presents a conceptual approach to personalising the selection of antibiotics to be tested for susceptibility, and a computational validation of this approach on a large UTI dataset from one of the Boston-based hospitals. At a high level, the authors analyse a range of patient demographic factors as input to a logistic regression model to determine probabilities of susceptibility for the standard antibiotic drugs. They then suggest that instead of testing a standard panel of 6 antibiotics it could instead be advantageous to test a personalised panel based on the model's predictions. The advantages would be to provide the clinician with a broader range of options associated with low resistance risk (more susceptible tests are expected for the Access category of antibiotics in the cohort receiving the personalised panel in simulation), and potentially to reduce the risk of side effects from using harsher second- and third-line antibiotics when they are not necessarily needed.

The analytical approach is reasonable and transparent, although I have not had the chance to run all the code in the repository because of the hurdles in the way of obtaining the required MIMIC dataset. I have several major and minor comments which I believe could substantially improve this manuscript.

Major:

1) The generalisability of the approach remains unclear because, despite following best protocols for train-test-validation splitting of the data, it has not been validated on an independent dataset, ideally from a different hospital system or even better, a different country (noting that the authors are based in the UK made me wonder if a similar dataset could be obtained for one the NHS trusts). If no other similar dataset can be obtained, at the very least a sensitivity analysis that should be done is to keep the last 2-3 years of the data for validation, and use only the earlier years to train and test the models. In this way any possible effect from increasing resistance trends could be mitigated.

2) The value of the approach is unproven; to do so would require demonstrating cost-effectiveness at a patient, hospital, or population level. As the exact relationship between antibiotic use and resistance levels is poorly understood, the authors could use data from, e.g., the GBD study to assess the expected severity of side effects of "watch" and "reserve" class antibiotics whose use could be avoided via the personalised panels, and trade it off against the estimated extra cost of personalised panels, to show that this could be cost-effective on a patient level as long as the personalisation is not too onerous. Without such an analysis, the approach is conceptually nice, but not as convincing.

Minor:

1) The codebase seems somewhat messy - to improve it I would suggest: a) improving the clarity of the names of the script files (1, 2, 3 is not as useful as saying what exact analysis type the script can perform) b) refactoring them to remove repetitiveness (in particular by encapsulating more of the code inside functions) and c) adding more comments throughout. Further separation into preprocessing, model training, evaluation, and graphical representation of the results, would also help with readability.

2) The statement that the results may not be identical when the code is rerun should be avoided by the use of a specific seed

for the random number generator hard-coded into a setup file at the start of the process. This ensures that the exact results can be obtained by running the code as is, and that variability can be assessed by changing the seed to a different one. Alternatively, the seed can be put into a configuration file outside of a script (e.g. in YAML format).

3) The decision to use the most recent urine specimen for each patient was not sufficiently justified; is the idea that they are more likely to have fewer available (susceptible) options due to previous treatment? As far as I can tell there is only a categorical variable for "previous treatment" included in the model; however, it may instead be more informative to include the number of previous visits for a UTI, or even to separate the patients into a "recurrent/chronic UTI" vs "first-time UTI" category. In either case, more justification should be provided.

Version 1:

Reviewer comments:

Reviewer #2

(Remarks to the Author)

My previous concerns have been addressed (apart from the potential benefits of a more specialised journal for this work).

Peer review team
Nature Communications

Reviewer #1 comments:

Fascinating study that applies a model to predict susceptibility results to WHO ACCESS drugs to promote their testing and reporting with the intention of increasing their use.

Very well written and clear manuscript. Do not have many specific edits, however would request the authors to comment on the following in the discussion in more clarity.

Reviewer comment	Author response
1. When choosing to apply the predication model at the start of the laboratory process, it seems to fail to select agents that despite being on the ACCESS list would be more clinically suitable for the organism or rather less suitable to use as it would be considered a organism specific drug eg. ceftazidime should be reserved Pseudomonas or meropenem would be reported when it should be reserved as 2nd or even 3rd line option. This could have the unintended consequences of driving AMR and losing these last line agents instead of driving ABX use to narrower spectrum options specific to the organism	Thank you for your helpful peer review comments. We agree that this is an additional piece of context that could be considered in the application of the algorithm. We have therefore added this consideration to the algorithmic rules that we have suggested as ways of incorporating the personalised approach with existing standard operating procedures in the discussion (lines 398-400).
2. The desire to report a longer list of susceptible antibiotics is not always the objective. this can inadvertently direct clinicians to choose agents that again are 2nd or third line agents.	This is a good point and is why we also examined the ability of the approach to provide at least one susceptible result. We have added some discussion around the relative merits of more susceptible results per panel (e.g., where antimicrobial contraindications exist) and providing at least one susceptible result per panel (lines 380-386).
3. It seems that it would be more helpful to help lab workflow and patient care to apply the prediction model at the time of organism identification. However, based on your findings the prediction outcome was less beneficial. Can you clarify what could be done to improve the prediction at this point in the laboratory workflow?	We have now added discussion points to describe how laboratory information management systems and standard operating procedures could be adapted alongside personalised AST to improve information availability, and therefore predictions early in the specimen pathway (lines 373-377)

Reviewer #2 comments:

This paper, although relatively simple analytically, is technically well executed and has the potential to inform current and future work attempting to improve efficiency and outcomes by predictive personalisation of clinical microbiology lab assays.

Reviewer comment	Author response
1. The results obtained are likely to of interest to largely clinical lab microbiologists/scientists rather than a broader audience. For this reason it may have greater impact in a more specialised clinical/medical microbiology journal.	Thank you for your constructive and insightful comments on our manuscript and codebase, it is much appreciated. We strongly believe that the results will be of interest to a broad audience – UTI is the world's commonest bacterial infection, and many patients will recognise the frustration of having to wait days for a result that informs safe antibiotic treatment, as will clinicians and healthcare managers who are expected to make healthcare decisions ever faster with ever fewer resources. The manuscript will be of particular interest in low- and middle-income settings where there is not the option to run large panels of antimicrobial susceptibility tests. Furthermore, we believe that the recognition of the importance of Access category antimicrobials in a forthcoming United Nations General Assembly, and the citing of diagnostics as a key World Health Organisation priority in AMR control, enhance the timeliness and importance of this work.
2. Line 70 (Introduction) - given the importance of appropriate use of interpretive criteria for MIC/Zol data this should probably at least mention them	Thank you, we agree and have added this point (lines 75-76).
3. Line 76 (Introduction) - which results are actually reported out may also vary	We have now added this point (lines 84-85).
4. Line 425 (Methods) - SIR under which set of guidelines? Presumably CLSI for US-derived data?	This information is not provided by the dataset, but yes based on US practice we believe CLSI interpretation to be likely – we have added a point to this effect (lines 490-492).
5. Line 437 (Methods) - Please provide citations and not just URL links to the key libraries used - these citations are an important metric that funds their on-going development! This is specifically outlined and requested by most of the libraries used in this paper e.g., https://scikit-learn.org/stable/about.html#citing-scikit-learn https://rasbt.github.io/mlxtend/ https://rstudio.github.io/reticulate/authors.html#citation https://numpy.org/citing-numpy/ etc	Thank you, we have now added the citations you have suggested plus citations for Pandas, AMR and Shiny.
6. Line 487 (Methods) - Should these details not come before the microsimulation study?	We agree that this would help flow and better delineate the clinical prediction model and the microsimulation study elements. We have therefore split the statistical methods section into clinical prediction model statistical methods (moved before the microsimulation study as you have suggested) and microsimulation study statistical methods (in the original place at the end of the manuscript).
7. Table 1 (Results) - Is the remaining % female or are there rows with no data?	The remaining % are female – we have added this to the table to clarify.
8. Table 2 (Results) - What was the variances of these metrics across CV folds?	This table reports only these point metrics on the single validation run that was performed using the final model on the validation dataset, so variances cannot be applied. We have, however, now increased the level of detail in the stability cross-validation analysis results (lines 156-164) to report the largest standard deviation observed in AUC-ROC when the size of the training dataset was reduced. We have also added an out-of-sample cross-validation analysis where training datasets from one time period are cross-validated against a range of

	other time periods, and we have reported the largest standard deviation in AUC-ROC observed in this analysis (lines 204-214).
9. Table 2 (Results) - How are metrics for both R and S reported? It is a bit unusual to report both for binary classification as it involves treating S as both a true positive and true negative in each calculation. Did you just flip the confusion matrix diagonals and recalculate? I can see the justification of providing more context but worth expanding upon a little bit due to its unusual nature.	Thank you for pointing this out – after some consideration, we agree that the benefit of leaving both class predictions in for context is beneficial, given the different clinical consequences of misidentifying susceptible and positive results in different AMR prevalence settings. We have added some analysis to the results section to contextualise this (lines 145-147) and explained in the methods that the metrics were calculated using the scikit-learn classification_report method (line 534-537).
10. Discussion - Another major consideration of adoption is the logistics of monitoring model performance over time and how retraining will be performed. Predictive models are generally legislated as medical device leading to considerable administrative overheads and challenges.	Thank you, this is an excellent point. We have added these considerations to the discussion (lines 411-415).
11. Highly recommend avoiding using globals in python instead of explicitly returning objects from functions - its very easy to introduce subtle bugs.	Thank you – we have tried to do this where possible, but given the number of models that needed to be run we enclosed as many of the operations as possible in functions that would add additional outcome measures to global dictionaries as the code runs and to help limit the length of the scripts.
12. Only import the libraries you are actually using in the code.	Thank you, we have now removed all unused libraries from the GitHub code.
13. Is the code for calculation of performance metrics presented in the paper included?	The code for performance metric calculation is in the LR_multi_final function in the 'Imports & functions.py' file in GitHub (https://github.com/amh312/PDAST/blob/main/Imports%20%26%20functions.py) – lines 578 for the classification report and line 589-601 for the ROC curve.

Reviewer #3 comments:

The analytical approach is reasonable and transparent, although I have not had the chance to run all the code in the repository because of the hurdles in the way of obtaining the required MIMIC dataset. I have several major and minor comments which I believe could substantially improve this manuscript.

Reviewer comment	Author response
1. The generalisability of the approach remains unclear because, despite following best protocols for train-test-validation splitting of the data, it has not been validated on an independent dataset, ideally from a different hospital system or even better, a different country (noting that the authors are based in the UK made me wonder if a similar dataset could be obtained for one the NHS trusts). If no other similar dataset can be obtained, at the very least a sensitivity analysis that should be done is to keep the last 2-3 years of the data for validation, and use only the earlier years to train and test the models. In this way any possible effect from increasing resistance trends could be mitigated.	Thank you for your useful and insightful comments, which we believe have given us the opportunity to significantly strengthen our article. We agree that being able to validate data out of sample is important to demonstrate generalisability of the approach. We cannot obtain another similar dataset, so we have followed your good suggestion to validate in another time frame (lines 204-214 in results, lines 539-548 in methods) – a particularly pertinent approach for AMR in view of its rapidly increasing prevalence. We have done this by splitting the data into four time windows and validated a random training dataset from each time window against a random testing dataset from itself and the other three time windows for each antimicrobial agent model, using 20 random train-test splits for each analysis (3,840 validation runs in total). We have then analysed the variation and spread of AUC-ROC values out of sample compared to in-sample and observed key trends. These results are also displayed graphically in Supplementary Figure 3, and we have provided information on the size of each group in Table 1. We have also expanded on our last point in limitations to state that the study was performed in a single area, and the need for future validation in a range of global settings (lines 430-438).
2. The value of the approach is unproven; to do so would require demonstrating cost-effectiveness at a patient, hospital, or population level. As the exact relationship between antibiotic use and resistance levels is poorly understood, the authors could use data from, e.g., the GBD study to assess the expected severity of side effects of "watch" and "reserve" class antibiotics whose use could be avoided via the personalised panels, and trade it off against the estimated extra cost of personalised panels, to show that this could be cost-effective on a patient level as long as the personalisation is not too onerous. Without such an analysis, the approach is conceptually nice, but not as convincing.	We agree that more could be done with the available data to understand the clinical context into which results would be released, with specific regard to better understanding how the value of personalised AST could be maximised. We have therefore added two additional analyses (lines 273-311 and Figure 6) to the microsimulation: firstly, an analysis of the number of results for antimicrobial agents that patients were prescribed at the time of the result, and secondly, the number of cases in which this would facilitate a swap from a Watch agent to an access agent, and an intravenous agent to an oral agent. We have also added some discussion around the findings of these analyses (387-392). We do not believe it is currently possible to robustly prove the value of the approach with available data using health economic methods and data from sources like the Global Burden of AMR study, for two reasons: firstly, no utility measure has yet been developed for antimicrobial resistance that fully accounts for its individual and population impact – it is partly for this reason that the AWaRe classification exists and has been adopted by the United Nations to inform targets for global AMR control; and secondly, it would require prediction of clinician behaviour in response to results. Any health economic analysis therefore risks giving misleading results with the tools currently available. We have added these issues to our limitations section (lines 420-430) and suggested examples of research and technology that is likely to be required to realise them.
3. The codebase seems somewhat messy - to improve it I would suggest: a) improving the clarity of the names of the script files (1, 2, 3 is not as useful as saying what exact analysis type the script can perform), b) refactoring them to remove repetitiveness (in particular by	Thank you, this has been a particularly helpful suggestion in conducting the further analyses required – we have done as you have advised changed the names of the script files to better described their roles/contents, extensively refactored the code and placed into functions (kept in a section at the top of each script in which they are used) wherever there was significant code repetition, and added more comments throughout for explanation.

encapsulating more of the code inside functions), c) adding more comments throughout.	The updated repository is located at https://github.com/amh312/PDAST/
4. Further separation into preprocessing, model training, evaluation, and graphical representation of the results, would also help with readability.	We have separated code into sections as advised and updated the GitHub README to explain the order in which to run each section.
5. The statement that the results may not be identical when the code is rerun should be avoided by the use of a specific seed for the random number generator hard coded into a setup file at the start of the process. This ensures that the exact results can be obtained by running the code as is, and that variability can be assessed by changing the seed to a different one. Alternatively, the seed can be put into a configuration file outside of a script (e.g. in YAML format).	Thank you for this suggestion, which has again been helpful in the process of code testing when refactoring. The code now specifies seeds at every point where randomisation occurs to ensure that results are reproducible, and we have removed this statement from the GitHub repository README.
6. The decision to use the most recent urine specimen for each patient was not sufficiently justified; is the idea that they are more likely to have fewer available (susceptible) options due to previous treatment? As far as I can tell there is only a categorical variable for "previous treatment" included in the model; however, it may instead be more informative to include the number of previous visits for a UTI, or even to separate the patients into a "recurrent/chronic UTI" vs "first-time UTI" category. In either case, more justification should be provided.	This is a good point – we took this step to ensure there was no cross-pollution of training, validation, and microsimulation datasets with the same patients, but appreciate that this may introduce loss of important information. We have therefore added a feature variable for a previous coded diagnosis of UTI to the model as you have advised and provided a justification for the approach in the methods (lines 475-477).
7. As I mentioned in the main review, the main hurdle for reproducing the results has been in gaining access to the dataset, which requires some training to be followed beforehand. I have been able to run the code on the simulated data provided after a bit of struggle with the installation, however.	We apologise for the inconvenience but are bound by the terms of MIMIC-IV dataset access. The online training is relatively short, but we appreciate that this is a lot to ask for peer review and are grateful for your patience in persevering with the CodeOcean codebase.

Yours sincerely,

Dr Alex Howard
Consultant in Medical Microbiology
University of Liverpool & Liverpool University Hospitals NHS Foundation Trust